# Influences of Cluster Thinning on Fatty Acids and Green Leaf Volatiles in the Production of Cabernet Sauvignon Grapes and Wines in the Northwest of China

**DOI:** 10.3390/plants13091225

**Published:** 2024-04-28

**Authors:** Xiaoyu Xu, Chifang Cheng, Xu Qian, Ying Shi, Changqing Duan, Yibin Lan

**Affiliations:** 1Centre for Viticulture & Enology, College of Food Science and Nutritional Engineering, China Agricultural University, Beijing 100083, China; xuxiaoyu62429@163.com (X.X.); shiy@cau.edu.cn (Y.S.); chqduan@cau.edu.cn (C.D.); 2Key Laboratory of Viticulture and Enology, Ministry of Agriculture and Rural Affairs, College of Food Science and Nutritional Engineering, China Agricultural University, Beijing 100083, China; 3Xinjiang Wine Industry Innovation Research Institute, Manasi 832200, China; chengchifang@citicguoanwine.com; 4School of Biology and Food Engineering, Changshu Institute of Technology, Changshu 215500, China; qianxu@cslg.edu.cn

**Keywords:** cluster thinning, LCFAs, C_6_ volatiles, LOX pathway, parcels, PLS-DA

## Abstract

Cluster thinning has been widely applied in yield management and its effect on green leaf volatiles (GLVs) in wines has seldom been studied. GLVs are important flavor compositions for grapes and wines. This work aimed to investigate the impact of cluster thinning on these volatiles and their precursors in grapes and wines. Severe cluster thinning (CT1) and medium cluster thinning (CT2) were performed on Cabernet Sauvignon (*Vitis vinifera* L.) vines in two sites (G-farm and Y-farm) from Xinjiang province in the Northwest of China. The impact of cluster thinning treatments on the accumulation of GLVs and their precursors, long chain fatty acids (LCFAs) of grape berries and C_6_ volatiles, in resulting wines was investigated. Multivariate analysis showed that cluster thinning treatments induced significant changes in fruit and wine composition in both farms. In Y-farm, medium cluster thinning (CT2) significantly increased the average cluster weight of harvested berries. Additionally, both cluster thinning treatments (CT1 and CT2) increased fatty acids in harvested berries and CT2 led to an increase in C_6_ esters and a decrease in C_6_ alcohols in the wines of Y-farm under the warmer and drier 2012 vintage. However, the effect of cluster thinning was likely negative in G-farm due to its wetter soil and excessive organic matter. The treatments may be applicable for local grape growers to improve viticultural practices for the more balanced vegetative and reproductive growth of Cabernet Sauvignon grapevines. This work also provided further knowledge on the regulation of fatty acids and the derived C_6_ volatiles through the lipoxygenase (LOX) pathway.

## 1. Introduction

Cluster thinning is a technique commonly used to balance the source–sink relationship of vines [1,2]. Various studies have explored the effect of crop thinning in several wine regions on different varieties. The timing [3,4,5]; the intensity [6,7]; the methods [8,9] of crop thinning; and the interaction with other treatments, such as irrigation [10,11] and canopy management [12,13], have been widely studied. Work by Wang et al. [5] reported that both early thinning (berries at pea size) and late thinning (at veraison) significantly increased pruning weight and decreased yield; however, leaf photosynthetic rate and anthocyanin accumulation in berries were inconsistent between the two thinning treatments in Cabernet Sauvignon grapes. Regarding the intensity of cluster thinning, it was reported that high-intensity thinning led to significantly more total soluble solids (TSS) in grape berries and pH and total extract content in wine [7]. There is no agreement on the influence of crop load on vine vegetative growth, berry components, and other compositions of grapes and wine among researchers in recent years. Several studies have demonstrated that cluster thinning advanced berry ripeness together with increasing sugars or TSS and lowering titratable acidity in several varieties, such as Cabernet Sauvignon [14], Sangiovese [15], Pinot noir [16], Chardonnay Musqué [17], and Shenhua [18]. However, cluster thinning had little or no influence on vegetative growth, fruit ripening, and composition in other studies [6,19]. Moreover, the role of cluster thinning may also be highly modulated by environmental conditions and vine water status [20,21].

Though some studies examined the effect of cluster thinning on sensory attributes of wines, the results reported from various studies are far from conclusive. Ough and Nagaoka observed no difference in aroma quality and intensity in Cabernet Sauvignon wines from Napa Valley made from grapes under different crop thinning levels; however, differences were found between wines from two locations [22]. It was documented that bunch thinning at mid-veraison led to yield reduction and an increase in the TSS of harvested berries and alcohol level of produced wines but did not affect the rotundone concentration of Duras wines from France [12]. Another bunch-thinning study performed on Tempranillo vines showed that wines from bunch-thinning grapes possessed a stronger herbaceous aroma and alcoholic sensation in taste in addition to deepened color and high acidity [11]. Diago, Vilanova, Blanco, and Tardaguila [9] reported that mechanical thinning decreased the intensity of the aroma of Tempranillo wines. In another study, Cabernet Sauvignon wines made from vines pruned to low bud numbers (24 nodes/vine) in the early season (winter) were high in herbaceous notes and a bell pepper aroma compared with ‘high-yield’ wines (48 nodes/vine); however, the influence of cluster thinning on sensory attributes was not as significant as that of preseason pruning [23]. Cluster thinning in the early stage of berry growth led to more TSS, pH, and potential volatile terpenes compared with control grapes for Chardonnay Musqué [24]. It is well known that any viticultural practice conducted during grape development, such as irrigation [25,26], fertilization [27], and defoliation [28], could potentially exert various influences on grapes, and hence the aroma profile of wines, by regulating the metabolic pathways of volatile accumulation.

Fatty acids in grapes are important nutritional components for yeast growth and metabolism and, also, the precursors of fermented odorant compounds in wine. Higher unsaturated fatty acids in the fermentation medium can promote the growth of yeast, resulting in increased production of volatile compounds in wine [29,30]. Fatty-acid-derived volatiles in grape berries are comprised of C_6_ and C_9_ alcohols, aldehydes, and esters. C_6_ volatile compounds, derived from long-chain fatty acids by the lipoxygenase-hydroperoxides lyase (LOX-HPL) pathway of oxylipins metabolism, were the most abundant volatiles in grapes [31]. They were also classified as green leaf volatiles (GLVs) due to the characteristic ‘green’ and ‘herbaceous’ odors [32]. Furthermore, the C_6_ compounds were reported to be direct precursors for the production of hexyl acetate in wines [33]. For example, the C_6_ acetates were reported to be the most important volatiles for varietal difference between Shiraz and Cabernet Sauvignon wines, which was positively related to the higher synthesis of C_6_ compounds in Shiraz grapes [34]. Additionally, there have been several studies demonstrating that the production of GLVs in plants is influenced by environmental conditions (rainfall, light, and temperature) [35] and vineyard management [36]. On the other hand, increased irrigation and additional nitrogen fertilization had no significant impact on the concentration of the C_6_ compounds of Merlot grapes and wines in California, USA [37]. With all the work mentioned above, the significance of C_6_ compounds for grape berries and wines has been well acknowledged and the aspects influencing the production of these volatiles in grapes need to be better elucidated through chemical analysis under purpose-driven field experiments. Thus, the effect of crop load via cluster thinning on these compounds displays great significance for grapes and wines.

In the context of Cabernet Sauvignon in Xinjiang China, the influence of crop loads on grape berry and wine volatile profiles is not well documented. The effect of crop thinning on green aroma or C_6_ volatiles has been seldom reported. To address this void, a two-year field study examined the effects of cluster thinning on the yield components and C_6_ volatiles of Cabernet Sauvignon. The experiment was conducted in two farms in two consecutive vintages. Long-chain fatty acids and their derived C_6_ volatile compounds were analyzed in samples collected during the grape development with different crop levels. Results from this work will expand the knowledge of the influence of crop thinning on C_6_ volatile compounds and their precursors in semi-arid continental climates.

## 2. Results and Discussion

### 2.1. Viticulture Parameters

#### 2.1.1. Yield Components

The cluster thinning treatments significantly reduced the cluster number per new shoot by approximately 40–56% (CT-1) and 19–32% (CT-2) compared to the control group (CT-3) (Table 1). The CT-1 and CT-2 treatments decreased the yield by approximately 56–65% and 23–32% in comparison to the CT-3 vines in G-farm. In Y-farm, CT-2 vines produced the highest yield, followed by CT-3 and CT-1, which can be explained by the over 48% increase in the average cluster weight of CT-2 in comparison with the control group (CT-3). Relatively, a slight change in average cluster weight among the three groups in G-farm was observed (Table 1).

Several authors have reported a decrease in the yield of vines subjected to cluster thinning regardless of the thinning time; whereas, no significant effect was observed on cluster weight [5,38,39]. On the contrary, it was also reported that cluster thinning mostly led to a compensatory increment in cluster weight, especially under dry-farmed treatment; although, it varied with the irrigation status and year [21]. In our work, a more obvious increment of cluster weight was observed in Y-farm, especially in those subjected to medium-level cluster thinning (CT-2), compared to G-farm, which may be attributed to the soil having lower water content in Y-farm than G-farm. This was consistent with previous findings [21].

#### 2.1.2. Fruit Composition

The influence of cluster thinning on berry weight, TSS, and titratable acidity in Cabernet Sauvignon grape berries varied according to vintages and sites (Appendix A). Comparing the differences in berry physiochemical parameters between the two seasons, berries were much heavier and had less TSS in 2011 than in 2012 in both farms. Rainfall during the berry development was much higher in 2011 than in 2012 (Appendix A) and the continuous rainfall was unfavorable for the accumulation of sugars in berries. For each year, lighter berry weight, fewer TSS, and slightly higher TA were observed in G-farm compared with Y-farm. This phenomenon may be attributed to the wetter soils and excessive organic matter in G-farm. With respect to the effects of cluster thinning on berry weight, it could be found that both CT-1 and CT-2 led to the increased berry weight in Y-farm, respectively, at 76 DAF in 2011 and 115 DAF (harvest) in 2012. However, cluster thinning had little or no significant impact on berry weight in G-farm.

In Y-farm, both CT-1 and CT-2 berries showed significantly more TSS and lower titratable acidity than CT-3 berries at 65 DAF in 2011; however, this effect vanished at harvest. This could be explained by the fact that berries of lower crop levels ripened faster than those of higher crop levels and berries from different crop levels reached similar TSS levels by delaying the harvest of high-crop-level fruit [21]. In contrast, TSS and titratable acidity were not significantly influenced by cluster thinning at any developmental stage in the same farm in 2012. The vintage differences in berry physiochemical parameters may be explained through climate variation during the growing season in Y-farm, as reported by Bowen and Reynolds [40]. In the warmer and drier 2012 vintage, the effect of cluster thinning was minimized because all three treatments reached optimal ripeness for harvesting and winemaking. Whereas, the reduction in crop stress in vines advanced grape maturation in the cooler 2011 vintage. However, the effect of cluster thinning was distinctly different in G-farm. In general, both thinned CT-1 and CT-2 berries showed less TSS and higher TA than CT-3, especially at about 68 DAF. This result tended to refute some previous studies [7,17]. But a recent study also found that both 33% and 50% cluster thinning treatments decreased TSS and increased TA in Sugrathirteen grapes [41]. It suggested that in G-farm, with wetter soils and excessive organic matter, the effect of cluster thinning on berry chemical composition was likely negative.

### 2.2. Accumulation of Long-Chain Fatty Acids (LCFAs) and Fatty-Acid-Derived Volatiles in Grape Berries

In this work, a total of thirteen LCFAs (C ≥ 12) were detected in grape samples, including four unsaturated fatty acids and nine saturated fatty acids. The major long-chain fatty acids detected in this work were linoleic acid, linolenic acid, palmitic acid, stearic acid, and oleic acid (Figure 1). Linoleic acid and linolenic acid were the most abundant fatty acids in grape berries, in agreement with earlier results [42,43,44]. During the development of grape berries, the fatty acids roughly decreased after around 44 DAF, similar to the result from Demir and Namli’s work [45]. The highest concentration of linolenic acid (486.7 mg/kg) was observed in CT-2 berries collected at 44 DAF of the 2011 vintage from Y-farm. The other four major fatty acids peaked at around 56 DAF, relatively later than linolenic acid.

Regarding volatiles derived from fatty acids, a total of eleven C_6_ compounds and two C_9_ compounds were considered. It was reported that C_6_ alcohols dominated during late berry development preceded by aldehydes, which suggested the usage of alcohol to aldehyde ratios to predict the timing of harvest for a balanced grape and wine aroma [46]. In this work, both major C_6_ aldehydes, hexanal and (*E*)-2-hexenal, showed a significant increase after veraison, followed by a decrease in the harvest samples of Y-farm (Figure 1), which was consistent with an earlier study [47]. Whereas, C_6_ aldehydes increased slowly after veraison and presented high concentrations toward late berry development in G-farm. The differences seen in this study in the production of C_6_ aldehydes could be due to the obviously different water status and organic matter content between the two farms. C_6_ alcohols detected in our samples presented an increasing pattern during the development of grapes. C_6_ alcohols are derived from C_6_ aldehydes by alcohol dehydrogenases in grapes [48]. Levels of cis-3-hexenyl acetate and hexyl acetate were high in the early berry samples and significantly decreased toward late berry development, which was in good agreement with an earlier study [47]. Additionally, higher acetate content was observed in 2011 than in 2012, which may be explained through climate variation during the growing season. These results offer a further understanding of the evolution and variation of C_6_ compounds under different farm and season conditions.

### 2.3. The Influence of the Intensity of Cluster Thinning on LCFAs and Their Derived Volatiles of Harvested Grape Berries

The cluster analysis was performed to differentiate the harvested grape samples according to the concentration of major long-chain fatty acids and their derived volatiles. From Figure 2, samples from the two farms were clustered in different groups indicating the importance of sites in affecting these compounds of harvested berries. CT-1 and CT-2 grapes from Y-farm were closely clustered together in both vintages and CT-3-treated berries were discriminated from the other two treatments. Regarding G-farm, CT-1 and CT-3 grapes were clustered together in 2011 while CT-2 and CT-3 grapes were clustered together in 2012. Then the PLS-DA analysis was performed by LCFAs and volatiles from separate farms in two vintages (Figure 3). The first and second principal components (PC1 and PC2) represented 37.1% and 23.8% of the total variance in Y-farm and 30.5% and 7.8% of the total variance in G-farm, respectively. In Y-farm, CT-3 treatment was separated from the CT1 and CT2 treatments and fatty acids, mainly linolenic acid, were the most important factor discriminating them. Regarding the volatiles involved, (*E*)-2-hexen-1-ol, hexanal, and (*E*, *Z*)-2, 6-nonadienal were the top three volatiles with a variable importance in projection (VIP) score over 0.5 (Figure 3b). Regarding G-farm, the three treatments were not clearly separated from each other (Figure 3c). Hexanoic acid, linoleic acid, (*E*)-2-hexen-1-ol, hexanal, and (*Z*)-2-hexen-1-ol were the main compounds discriminating the three treatments (Figure 3d).

The variations in the content of major C_6_/C_9_ compounds among three treatments in two plots over the span of two years are illustrated in Figure 4. In Y-farm, the total concentration of C_6_ aldehydes was the highest in CT-1 and CT-2 berries of 2011 and in CT-3 berries of 2012. CT-3 berries contained the highest level of C_6_ alcohols in 2011 while the highest level was observed in CT-1 berries in the 2012 vintage. Only slight differences were found for C_9_ compounds, which were detected at relatively low levels. Regarding long-chain fatty acids, the lowest sum of fatty acids was observed in CT-3 berries in both vintages from Y-farm, suggesting that cluster thinning, at a medium or severe level, increased fatty acids in harvested berries. There has been no report about the influence of cluster thinning on the concentration of fatty acids in grapes yet. In general, higher levels of C_6_ aldehydes and alcohols, accompanied by less TSS and slightly higher TA in G-farm, implied a less ripened fruit in G-farm than in Y-farm. Regarding G-farm, as discussed above, the effect of cluster thinning was likely negative due to the wetter soils and excessive organic matter so its influence on the production of long-chain fatty acids, C_6_, and C_9_ volatiles may be significantly different from Y-farm. As seen in Figure 4, the lowest levels of C_6_ aldehydes, C_9_ volatiles, and fatty acids were observed in CT-2 berries and both cluster thinning treatments (CT-1 and CT-2) decreased the sum of long-chain fatty acids in the harvested berries of 2011 in G-farm. In another vintage, the CT-1 berry contained the lowest C_6_ and C_9_ volatiles.

A summary of the three-way ANOVA for each grape component and effect tested is displayed in Table 2. The treatment, vintage, farm, and their interactions were tested as sources of variation. Clustering thinning treatments showed an effect on 10 variables of the 16 volatiles measured. It was observed that all three factors (treatment, vintage, and farm) had a primary effect on the levels of (*E*)-2-Hexenal, 1-hexanol, (*E*)-2-hexen-1-ol, hexanoic acid, and (*E*)-2-nonenal; whereas, hexanal, ethyl hexanoate, hexyl acetate, and cis-3-hexenyl acetate were not significantly influenced by clustering thinning. Among fatty acids, linolenic acid, palmitic acid, and stearic acid were significantly affected by all three factors (treatment, vintage, and farm) while oleic acid and linoleic acid displayed no significant influence by cluster thinning treatments.

### 2.4. The Effects of Cluster Thinning Treatments on C_6_ and C_9_ Volatile Compounds in Wines

The concentrations of major volatile compounds (ethyl hexanoate, C_6_ alcohols, and C_9_ compounds) are displayed in Figure 4. Ethyl hexanoate was the most abundant C_6_ ester, which was massively produced during fermentation by the enzymes of yeast. According to previous reports in the literature [49], ethyl hexanoate contributes to red-berry aromas together with ethyl esters, such as ethyl caprylate, ethyl butyrate, and ethyl 3-hydroxybutyrate. CT-2 wines from Y-farm in 2012 presented the highest ethyl hexanoate content in comparison with CT-1 and CT-3 wines. Meanwhile, no difference was observed among the three treatments in G-farm. CT-2 wines from Y-farm also contained the lowest sum of C_6_ alcohols in 2012. On the contrary, CT-1 wines from G-farm in both vintages contained the lowest C_6_ alcohol levels. The level of C_9_ volatiles in all the wines was relatively low and the difference among different groups was not significant. By comparing the C_6_ volatiles from 2011 and 2012 vintages, it was found that wines from 2011 contained more abundant alcohols and the concentrations of esters were higher in 2012 wines. Considering the climatic difference between the two vintages, relatively heavier rainfall and lower sunlight in August of 2011 was observed in comparison with the same period of 2012. Our result was in complete agreement with our previous study [36]. In general, CT-2 treatment (medium thinning intensity) in Y-farm led to a higher concentration of the total amount of esters (ethyl hexanoate, hexyl acetate, ethyl 3-hexenoate, propyl hexanoate, ethyl 2-hexenoate, and ethyl 8-nonenoate; Appendix A) in the wines of 2012 along with a lower level of C_6_ alcohols (Figure 4), which could provide more favorable flavors to wines. From this point of view, CT-2 (medium thinning intensity) was more suitable for Cabernet Sauvignon grapes and wine in Y-farm. On the other side, CT-1 treatment produced fewer C_6_ alcohols and no significant differences in esters, which could be influential to the wine aromatic quality in G-farm. Thus, CT-1 treatment (high thinning intensity) would be the premium choice for G-farm considering the C_6_ volatiles of wines.

A three-way ANOVA was used to test the effects of the treatment, vintage, farm, and their interactions on the wine volatiles (Table 3). Among thirteen volatiles of wines in this work, five of them (ethyl hexanoate, ethyl 3-hexenoate, (*E*)-3-hexen-1-ol, ethyl 8-nonenoate, and hexanoic acid) were not significantly influenced by treatments. According to the result of this work, clustering thinning treatments significantly influenced the concentration of hexyl acetate, ethyl 2-hexenoate, (*Z*)-3-hexen-1-ol, (*Z*)-3-nonen-1-ol, and (*E, Z*)-2,6-nonadienal in the resulting wines. In addition, it can be seen that the treatment, farm, and vintage may exert a comprehensive impact on C_6_ and C_9_ volatiles in the resulting wines.

Regarding the influence of cluster thinning treatments, various conclusions can be made from previous reports. In some reports, cluster thinning has been shown to have a limited effect on the sensory properties of wines [9,17,50,51]. In other research, both early and late mechanical crop thinning were reported to reduce the total intensity of aroma and the herbaceous aroma of Tempranillo wines; however, crop thinning had no significant influence on the sensory attributes of Grenache wines [9]. However, bunch-thinning wines decreased the herbaceous descriptor and increased the intensity of the floral and red fruit aroma of Tempranillo under regular irrigation [11,24]. In the present work, medium cluster thinning (CT2) treatment led to an increase in C_6_ esters and a decrease in C_6_ alcohols in Y-farm in 2012; thus, it had the potential to increase the fruit aroma and decrease the green or grassy aroma of resulting wines.

## 3. Materials and Methods

### 3.1. Reagents and Standards

The internal standard (4-methyl-2-pentanol) and C_6_-C_24_ *n*-alkanes used for the identification and quantification of volatile compounds were purchased from Sigma-Aldrich (St. Louis, MO, USA). The methyl ester of long-chain fatty acids was purchased from Sigma-Aldrich and Fluka (Buchs, Switzerland). Other chemicals (NaOH, NaCl, menthol, dichloromethane, H_2_SO_4_, and hexane) were purchased from Beijing Chemical Works (Beijing, China).

### 3.2. Field Trials and Sample Collection

The experiments were conducted on Cabernet Sauvignon vines in two separate locations at Manas County (86°12′2″ E and 44°17′55″ N) of Xinjiang province, China, over two consecutive vintages, 2011 and 2012. This region has a semi-arid climate with low annual rainfall, a high effective daytime temperature, and a big daytime-to-nighttime temperature difference. All the meteorological data of 2011 and 2012 reported in a previous study from our research center have been listed in Appendix A [52]. With regard to the berry development period, growing degree days (GDDs) and sunlight duration in 2012 were higher than in 2011 and the total rainfall in 2012 was lower than in 2011 [52]. The distance between Y-farm and G-farm is 5 km. Both farms adhere to identical standards for water, fertilizer, and pest management to the winery site. Despite both farms being situated in a mesothermal climate, their soil compositions vary (Appendix A). Briefly, Y-farm soil contains higher sand and pH levels than G-farm; however, the organic matter and water content of soils are significantly higher in G-farm. These vines were own-rooted and were planted in 2000. The vines were arranged in north–south rows with a 2.5 × 1.0 m space and trained to a sloping trunk with a vertical shooting positioning trellis system (M-VSP). Each spur-pruned cordon had a bud load of 12–15 nodes per linear meter. For the trial, selected vines were randomly assigned to 3 levels of cluster thinning treatments, including CT-1 (1 cluster per shoot thinned, 1 cluster/shoot reserved), CT-2 (1 cluster every other shoot thinned, average 1.5 clusters/shoot reserved), and CT-3 (not thinned, 2 clusters/shoot reserved) as the control group (Appendix A). CT-1 was considered a severe cluster thinning treatment and CT-2 was a medium cluster thinning treatment, in relative terms. A randomized complete block design was employed, with 3 replicates and each replicate consisting of 15 vines with a uniform growth state. Different numbers of clusters were removed manually at 26 days after flowering (DAF). Cluster thinning treatments were applied on the same vines for two consecutive vintages.

Grape samples were collected starting from 6 weeks after flowering till harvest at 44 DAF, 59 DAF (early-veraison), 65 DAF, 76 DAF, 93 DAF, and 114 DAF (berries harvest-ripe) for Y-farm in 2011; 45 DAF, 59 DAF (early-veraison), 69 DAF, 73 DAF, 84 DAF, 99 DAF, and 115 DAF (berries harvest-ripe) for G-farm in 2011; and 46 DAF, 56 DAF (early-veraison), 68 DAF, 75 DAF, 89 DAF, 103 DAF, and 115 DAF (berries harvest-ripe) for Y-farm and G-farm in 2012. Grape berries were sampled from 45 vines from 3 rows (15 vines per row) for each treatment to provide 3 biological replicates. For each berry development period, 300 berries were collected for each replicate and they were immediately frozen in liquid nitrogen and stored at −80 °C for further analysis. Total soluble solids (TSS) were measured with a PAL-1 Digital Hand-held “Pocket” Refractometer (Atago, Tokyo, Japan). Titratable acidity (TA) was determined by titration with NaOH to the end point of pH 8.2 and expressed as tartaric acid equivalent.

### 3.3. GC-MS Analysis of C_6_ Volatile Compounds in Grapes and Wines

The volatile compounds of grape berries and wines were extracted and analyzed according to an available method developed by our research center [53,54]. Briefly, seeds were removed from the berry before extraction. The remained berries were ground and blended with 1 g of polyvinylpolypyrrolidone (PVPP) under continuous liquid nitrogen. The mixture was macerated for 4 h (4 °C) and centrifuged at 7104× *g* (8000 rpm, 10 cm) at 4 °C for 15 min to obtain clear juice. In total, 5 milliliters of clear juice (or wine sample) with 10 µL of internal standard 4-methyl-2-pentanol (4M2P, 1.0038 g/L) and 1 g of NaCl were added to a 15 mL sample vial tightly capped with the PTFE-silicon septum containing a magnetic stirrer. The vial was equilibrated at 40 °C for 30 min. Then, the pretreated SPME fiber (50/30-μm DVB/Carboxen/PDMS, Supelco, Bellefonte, PA, USA) was inserted into the headspace of the sample and volatiles were extracted for 30 min. After that, the fiber was immediately desorbed for 8 min in the GC injector. The analysis of volatiles was performed on an Agilent 6890 GC equipped with Agilent 5975 MS fitted with a 60 m × 0.25 mm HP-INNOWAX capillary column with 0.25 μm film thickness (J&W Scientific, Folsom, CA, USA). The oven temperature was programmed as follows: 50 °C for 1 min, increased to 220 °C at the rate of 3 °C/min, and held at 220 °C for 5 min. The flow rate of helium was 1 mL/min. The ion source temperature was 250 °C. Electron ionization mass spectrometric data from *m*/*z* 30–350 u were collected. The ionization voltage was 70 eV. Two independent extractions were carried out for each sample.

The limits of detection (LODs) and quantification (LOQs) were calculated by MSD ChemStation Data Analysis (Agilent Technologies, Inc., Santa Clara, CA, USA) for the typical sig-nal-to-noise (S/N) ratios of 3 and 10, respectively. Aroma compounds were qualitatively assessed by matching the mass spectrum and retention index with the corresponding standards and the NIST 08 MS library. According to the sugar and total acid content in the samples, a grape synthetic solution was prepared in distilled water containing 5 g/L tartaric acid and 230 g/L glucose with pH adjusted to 3.8 with a 5 mol/L NaOH solution. Similarly, a wine synthetic solution contained 14% vol ethanol and 5 g/L tartaric acid and the pH was adjusted to 3.8. The standard solution was successively diluted into thirteen levels using the synthetic solution for the quantitative analysis of aroma compounds using the calibration curve of the aroma standard. Volatile compounds of each level were extracted and analyzed using the same method as the grape samples.

### 3.4. Analysis of Long-Chain Fatty Acids (LCFAs) in Grape Berries

The analysis method was based on previous reports with modifications described as follows [55,56]. The seed-off grape berries were deeply frozen in liquid nitrogen and then immediately pulverized. The powder was dried by vacuum freeze-drying until the moisture rate was lower than 5%. In total, 1 gram of dried powder was added into a 200 mL glass flask with 30 mL n-hexane as the extraction solvent. The mixture was extracted by ultrasonic technology under 28 °C for 30 min and the extraction was repeated twice. The organic phases were collected and dried at 30 °C by a vacuum rotary evaporator. Then, 5 mL 1% H_2_SO_4_/methanol (*w*/*v*) solution was added to methylate LCFAs at 65 °C for 2 h. After cooling to room temperature, 3 mL hexane and 3 mL distilled water were added to separate methyl esters of fatty acids (FAMEs). The separation was repeated three times. The combined hexane extract was concentrated under a gentle stream of nitrogen to 1 mL. Methyl nonadecanoate (0.4 mg/mL) was added as an internal standard. Then, the samples were analyzed by GC-MS immediately. The analysis of FAMEs was performed on an Agilent 6890 GC equipped with a 5975B MS system. The capillary column was a 60 m × 0.25 mm HP-INNOWAX capillary column with 0.25 μm film thickness (J&W Scientific, Folsom, CA, USA). One microliter of the extract was injected. The oven temperature was held at 80 °C for 1 min before being raised to 220 °C at 25 °C/min and, then, raised to 250 °C at 5 °C/min and held at 250 °C for 20 min. The temperature of the injector was 250 °C.

### 3.5. Vinification and Wine Sampling

Approximately 60 kg of Cabernet Sauvignon grapes of each cluster thinning treatment was harvested at commercial maturity for winemaking. The manually destemmed and crushed grapes were transferred into a 20 L glass container together with 1 mL pectinase (Lallzyme Ex, Lallemand, France) and 10 mL 6% H_2_SO_3_. They were then inoculated with activated Lalvin D254 commercial yeast according to the instructions. Alcoholic fermentation of must from three treatments was kept in the same condition and caps were submerged three times every day while temperature and density were monitored. At the end of the alcoholic fermentation, samples were collected and stored at −20 °C until analysis. The wine parameters of berries with different treatments from two farms in two vintages are listed in Appendix A.

### 3.6. Statistical Analysis

One-way ANOVA analysis and three-factor ANOVA analysis were performed in R (3.1.0). Hierarchical cluster analysis and partial least squares discriminant analysis (PLS-DA) were performed on MetaboAnalyst 3.0 and auto-scaling was used in data normalization [57]. Bar graphs and heatmaps were accomplished using the ‘ggplot2’ and ‘pheatmap’ packages in R (3.1.0), respectively. Pearson’s correlation analysis was performed by SPSS 20.0 for Windows (SPSS Inc., Chicago, IL, USA).

## 4. Conclusions

Two levels of manual cluster thinning treatments were performed on grape vines in two farms in the Northwest of China. Cluster thinning affected both the yield components of grapes and the level of linolenic acid, as well as green leaf volatiles, which are important for the aromatic quality of Cabernet Sauvignon grapes and wines. In Y-farm, medium cluster thinning (CT2) significantly increased the average cluster weight of harvested berries. Additionally, both cluster thinning treatments (CT1 and CT2) increased fatty acids in harvested berries and CT2 resulted in an increase in C_6_ esters and a decrease in C_6_ alcohols in wines of Y-farm under the warmer and drier 2012 vintage. However, the effect of cluster thinning on the viticulture parameters, the production of fatty acids, and C_6_ and C_9_ volatiles in grapes and wines were likely negative in G-farm, presumably caused by the wetter soils and excessive organic matter. This work offers insights into regulating yield in over-loaded vineyards from Xinjiang and provides further information on the regulation of fatty acids and the derived C_6_ and C_9_ volatiles through the LOX pathway. The results obtained may also be applicable for local grape growers to improve the balance of the vegetative and reproductive growth of Cabernet Sauvignon grapevines in order to produce premium wines.

## Figures and Tables

**Figure 1 plants-13-01225-f001:**
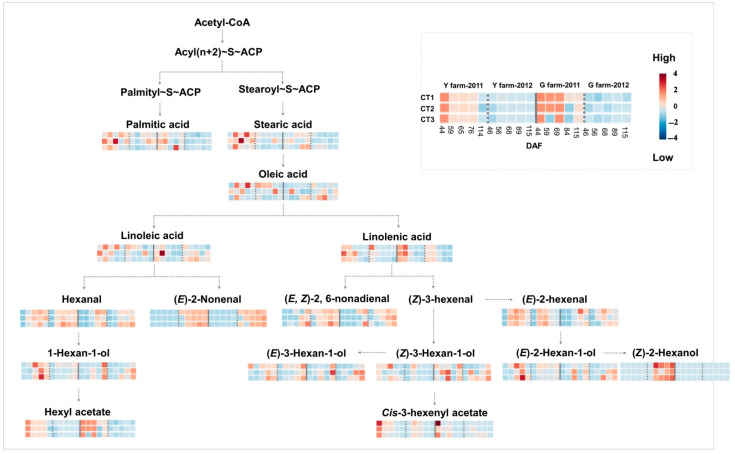
A brief figure of the pathway of synthesis of major long-chain fatty acids, C_6_, and C_9_ volatiles through the LOX-HPL route. Colored boxes indicating concentration variations of all samples during grape development from two farms in two vintages. Red indicates higher concentration and blue indicates lower concentration, relatively.

**Figure 2 plants-13-01225-f002:**
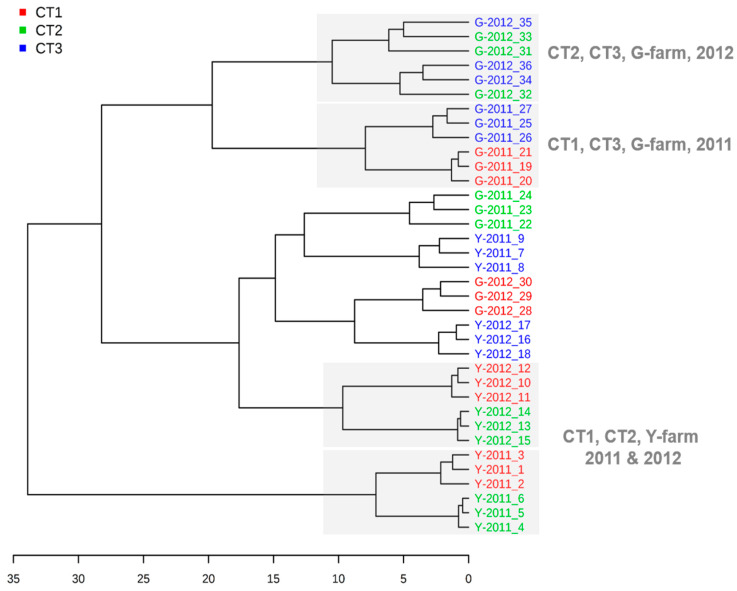
Cluster analysis with the concentration of major fatty acids and derived volatiles.

**Figure 3 plants-13-01225-f003:**
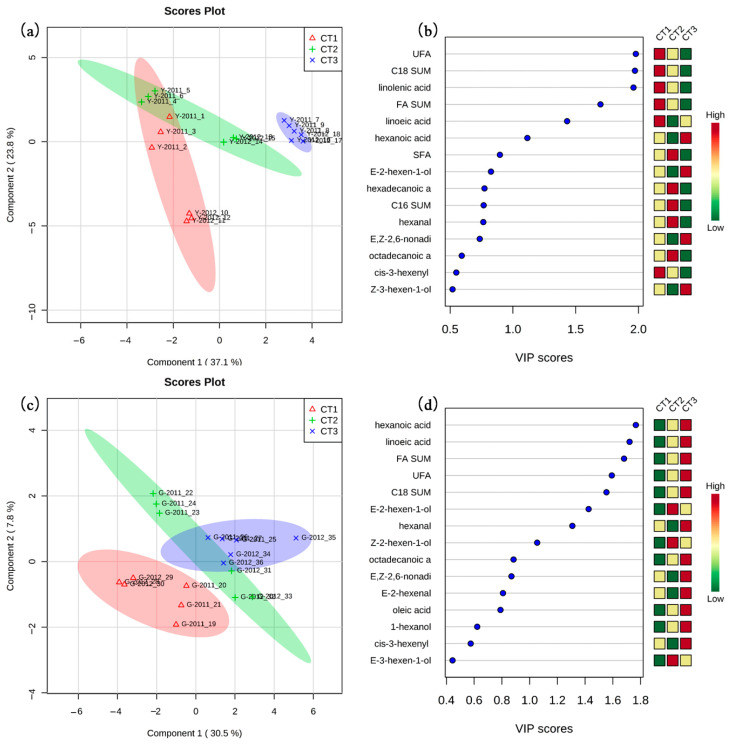
PLS-DA analysis of fatty acids and derived volatiles in harvested berries from two farms in two vintages. (**a**,**c**) Score plot of Y-farm and G-farm, respectively, and (**b**,**d**) selected compounds based on variable importance in projection (VIP) scores of Y-farm and G-farm, respectively. SFA, UFA, FA SUM, C_16_ SUM, and C_18_ SUM represent the sum of saturated fatty acids, unsaturated fatty acids, fatty acids, C_16_ fatty acids, and C_18_ fatty acids, respectively.

**Figure 4 plants-13-01225-f004:**
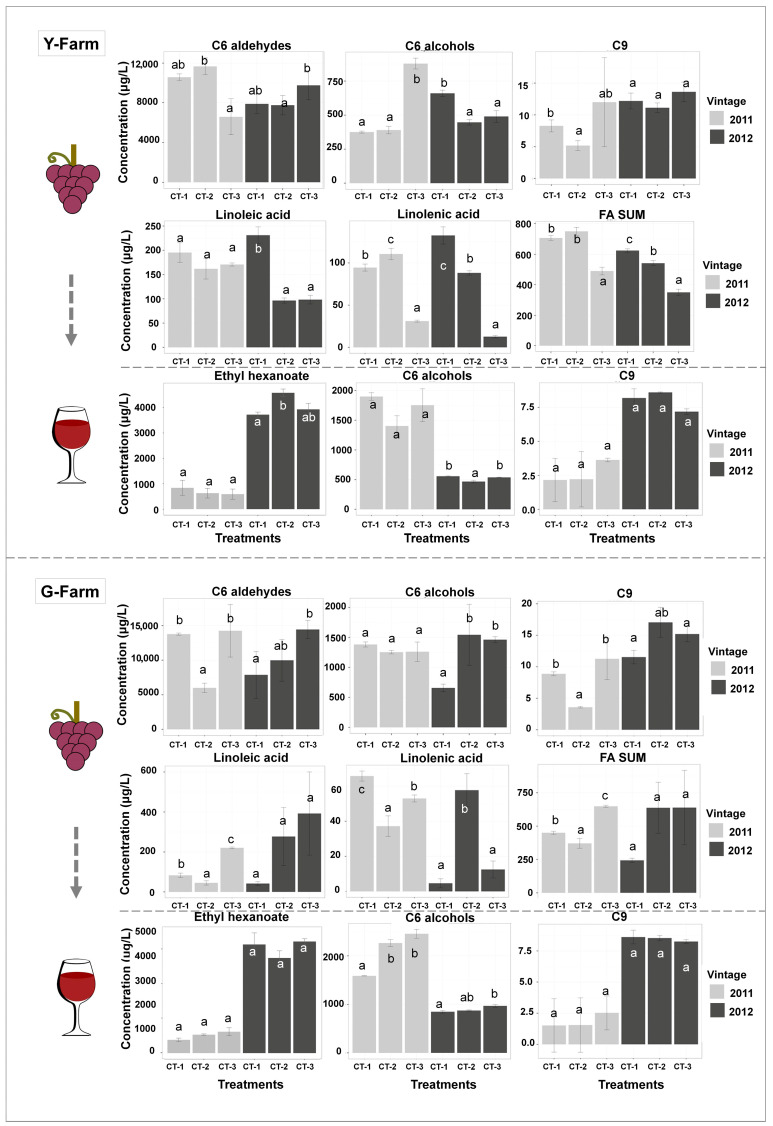
Bar plots of major compounds of harvested grapes and wines after alcoholic fermentation. Different letters mean significant differences in the concentration of these compounds among treatments in each vintage, according to the Duncan test (*p* < 0.05). FA SUM represents the sum of fatty acids.

**Table 1 plants-13-01225-t001:** Yield components of cluster thinning treatments in Y-farm and G-farm of the 2011 and 2012 vintages.

Treatment	Y-Farm			Treatment	G-Farm		
	Cluster Number/ New Shoot	Yield (t/hectare)	Average Cluster Weight (g)		Cluster Number/ New Shoot	Yield (t/hectare)	Average Cluster Weight (g)
2011-CT-1	1.03 ^a^	6.71 ^a^	120.52 ^a^	2011-CT-1	0.72 ^a^	5.37 ^a^	126.03 ^a^
2011-CT-2	1.37 ^ab^	12.76 ^b^	169.87 ^b^	2011-CT-2	1.12 ^a^	10.35 ^b^	147.81 ^b^
2011-CT-3	1.71 ^b^	10.71 ^b^	108.36 ^a^	2011-CT-3	1.64 ^b^	15.39 ^c^	146.52 ^b^
Significance	*	*	*	Significance	*	*	*
2012-CT-1	1.02 ^a^	6.14 ^a^	110.33 ^ab^	2012-CT-1	0.73 ^a^	4.46 ^a^	104.65
2012-CT-2	1.37 ^ab^	10.06 ^b^	133.94 ^b^	2012-CT-2	1.12 ^a^	7.76 ^ab^	110.9
2012-CT-3	1.71 ^b^	8.94 ^ab^	90.54 ^a^	2012-CT-3	1.65 ^b^	10.14 ^b^	96.6
Significance	*	*	*	Significance	*	*	ns

Different letters represent significant differences among treatments in each vintage according to the Duncan test (*: significant at *p* < 0.05). Cluster thinning treatments: CT-1, one cluster per shoot thinned, one cluster/shoot reserved; CT-2, one cluster every other shoot thinned, average one-and-a-half clusters/shoot reserved; CT-3, not thinned, two clusters/shoot reserved.

**Table 2 plants-13-01225-t002:** The effects of treatment, vintage, farm, and their interactions on the grape components by a three-way ANOVA.

Grape Components	Treatment	Farm	Vintage	Treatment × Farm	Treatment × Vintage	Farm × Vintage	Treatment × Farm × Vintage
Hexanal	ns	ns	ns	***	***	ns	**
(*E*)-2-Hexenal	**	***	**	***	**	ns	**
1-Hexanol	*	***	***	**	**	*	***
(*E*)-2-hexen-1-ol	***	***	**	***	*	ns	***
(*Z*)-2-hexen-1-ol	***	ns	***	***	***	ns	***
(*E*)-3-hexen-1-ol	**	***	ns	***	**	***	***
(*Z*)-3-hexen-1-ol	**	***	ns	ns	***	ns	***
Ethyl hexanoate	ns	ns	***	ns	ns	***	**
Hexyl acetate	ns	ns	ns	ns	ns	*	ns
*Cis*-3-Hexenyl acetate	ns	***	ns	ns	ns	ns	ns
Hexanoic acid	***	*	***	***	**	ns	***
(*E*)-2-nonenal	***	**	***	***	*	*	*
(*E*, *Z*)-2,6-nonadienal	**	ns	ns	ns	**	ns	ns
Palmitic acid	**	***	**	*	ns	ns	ns
Stearic acid	**	*	***	***	ns	ns	*
Oleic acid	ns	ns	ns	ns	ns	ns	ns
Linoleic acid	ns	ns	ns	***	ns	*	ns
Linolenic acid	***	***	***	***	***	***	***

ns, *, **, ***: not significant or significant at *p* < 0.05, 0.01 and 0.001, respectively.

**Table 3 plants-13-01225-t003:** The effects of the treatment, vintage, farm, and their interactions on the wine volatiles by a three-way ANOVA.

Wine Volatiles	Treatment	Farm	Vintage	Treatment × Farm	Treatment × Vintage	Farm × Vintage	Treatment × Farm × Vintage
Ethyl hexanoate	ns	ns	***	*	ns	ns	**
Hexyl acetate	***	***	***	***	***	***	***
Ethyl 3-hexenoate	ns	***	***	*	ns	***	*
Propyl hexanoate	*	***	***	***	*	***	***
Ethyl 2-hexenoate	**	ns	***	***	ns	**	ns
1-Hexanol	*	***	***	***	**	**	***
(*E*)-3-Hexen-1-ol	ns	***	***	***	**	***	**
(*Z*)-3-Hexen-1-ol	***	***	***	***	***	***	***
Ethyl 8-nonenoate	ns	ns	**	ns	ns	ns	ns
(*Z*)-3-Nonen-1-ol	**	*	***	**	**	*	**
(*E*)-6-Nonen-1-ol	*	***	***	ns	ns	***	ns
(*E*, *Z*)-2,6-Nonadienal	***	ns	***	**	***	ns	**
Hexanoic acid	ns	ns	***	ns	ns	ns	ns

ns, *, **, ***: not significant or significant at *p* < 0.05, 0.01, and 0.001, respectively.

## Data Availability

Data are contained within the article and Appendix A.

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
