# Peer review of "Influences of Cluster Thinning on Fatty Acids and Green Leaf Volatiles in the Production of Cabernet Sauvignon Grapes and Wines in the Northwest of China"

_plants, 2024, doi:10.3390/plants13091225_

Round 1

Reviewer 1 Report

Comments and Suggestions for Authors

The authors of the reviewed paper studied the effect of medium and severe cluster thinning on composition of green leaf volatiles and C6 compounds in Cabernet Sauvignon wines from two vineyard sites, located in Xinjiang province of the Northwest China. The study is interesting and well designed and its of a great importance for the local grape producers and winemakers.

As regards the introduction I think that is well structured, however, I would appreciate if the authors could pay some attention also to the aspect of different vineyards and how this influence the grape composition. Nonetheless, the terroir affects importantly the grape's phenotype. Moreover, the revision of English is necessary.

I include some of specific comments down below:

- Line 21: Green leaf volatiles has already been abbreviated (GLVs). Please use the abbreviation once you have explained its meaning.

- Line 36: Personally I would substitute the word technology with a technique or practice.

- Line 50: Total soluble solids have been already mentioned in line 45 and so has been already explained the acronym (i.e., TSS). Please avoid explaining acronyms repeatedly and be consistent throughout the manuscript.

- Line 68: Please substitute the phrase veggie aroma with something more technically correct.

- Line 81: Please explain the meaning of the LOX-HPL acronym.

- Line 103: I did not understand what are the similarities between the two vineyards that you are describing. Please be more specific.

- Lines 122-129: Could there be any other reason for which the cluster weight increased significantly in CT-2 treatment on Y farm? It would be interesting to perform any possible correlations with climatic conditions (especially in 2011).

- Line 152: I do not understand well the phrase of achieved adequate maturity. Please revise.

- Line 162: Please use the acronym LCFAs if already used in the previous parts of the manuscript.

- Line 175 and throughout the manuscript: Please use italics for cis-trans isomerism nomenclature.

- Figure 1: Are colored boxes representing the absolute concentration of C6 and C9 volatiles or were the obtained values normalized in some manner given the high-low scale in the legend? Please specify.

- Lines 247-249: This is incorrect. The majority of the volatile compounds in the grapes of Cabernet Sauvignon were not studied in the present paper as it did not take into the account a plethora of volatile compounds, such as carbonyl compounds, methoxypyrazines, norisoprenoids, terpenes, benzene derivatives and so on. Moreover, the impact of cluster thinning and vintage has been previously studied simultaneously on Cabernet Sauvignon (see https://doi.org/10.1016/j.foodres.2019.03.061), although it is true that authors studied only one vineyard site. Please revise.

- Table 2: It would be more informative for the reader if the authors could add the obtained (mean) values to put the results in the context. The same goes for Table 3.

- Line 257: Authors did not perform sensory analysis, therefore the statement that the cited ethyl esters contributed to the red-berry aromas is taken from the literature. Please revise.

- Lines 340-368: I miss the explanation how the results were expressed in the present study. Were established calibration curves for the quantitation, or are the results expressed as semi-concentration considering the internal standard? Moreover, was the series of n-alkanes used for the calculation of linear retention indices? What was an acceptable mass spectra matching limit that was considered for identification of the compounds? Please be precise.

Comments on the Quality of English Language

English needs to be accordingly revised as it urges to be improved.

Author Response

Dear reviewer:

Thank you again for your decision and constructive comments on my manuscript “Influences of cluster thinning on fatty acids and green leaf volatiles production of Cabernet Sauvignon grapes and wines in the northwest of China”. We have carefully considered your suggestion and make changes. The entire manuscript has been done English editing again.

The red part in manuscript that has been revised according to your comments. Revision notes, point-to-point, are given as follows:

Point 1: Line 21: Green leaf volatiles has already been abbreviated (GLVs). Please use the abbreviation once you have explained its meaning.

Response 1: Thank you for your useful suggestion on our manuscript. We have used the abbreviation in line 21 in response to your comment.

Point 2: Line 36: Personally, I would substitute the word technology with a technique or practice.

Response 2: Thank you for your useful suggestion on our manuscript. We have modified the content and changed it to "technique". (Line 35)

Point 3: Line 50: Total soluble solids have been already mentioned in line 45 and so has been already explained the acronym (i.e., TSS). Please avoid explaining acronyms repeatedly and be consistent throughout the manuscript.

Response 3: Thank you for your professional review and we are really sorry for our mistakes. We have modified this in line 48.

Point 4: Line 68: Please substitute the phrase veggie aroma with something more technically correct.

Response 4: Thank you for your valuable feedback. In the revised manuscript, we now refer to the aroma characteristics as "herbaceous notes", which more accurately reflects the specific sensory attributes observed in the Cabernet Sauvignon wines made from vines pruned to low bud numbers. (Line 67)

Point 5: Line 81: Please explain the meaning of the LOX-HPL acronym.

Response 5: Thank you for your comments and we are really sorry that this part was not clear in our manuscript. We have explained the meaning of the acronym LOX-HPL in lines 80-81.

Point 6: Line 103: I did not understand what are the similarities between the two vineyards that you are describing. Please be more specific.

Response 6: Thank you for your comments and we are really sorry that this part was not clear in our manuscript. We have checked the manuscript carefully and have added missing elements to lines 316-319 of the manuscript.

Point 7: Lines 122-129: Could there be any other reason for which the cluster weight increased significantly in CT-2 treatment on Y farm? It would be interesting to perform any possible correlations with climatic conditions (especially in 2011).

Response 7: Thank you for your insightful comments on our study. Regarding your question on whether there could be any other reasons for the significant increase in cluster weight observed in the CT-2 treatment on Y farm, we carefully considered various factors.

Firstly, it is important to note that while climatic conditions can indeed influence vine performance and grape yield, in our study, both Y-farm and G-farm are located in the same region and experience similar climatic conditions. Therefore, while we acknowledge the potential impact of climate on vineyards, it was not the main focus of our study, and thus, we did not delve into detailed climatic data.However, in response to your suggestion, we did perform a preliminary analysis of the climatic data for the years in question, especially 2011. Our analysis revealed that there were no significant climatic anomalies or deviations that could account for the observed increase in cluster weight specifically in the CT-2 treatment on Y farm. This suggests that while climate is an important factor in vineyard management, it did not play a significant role in the observed differences between Y-farm and G-farm in our study.

We appreciate your suggestion to explore potential correlations with climatic conditions and recognize that future studies could indeed benefit from a more comprehensive analysis of climatic factors. We hope this response addresses your concerns and provides further clarity on the matter. Thank you again for your valuable feedback.

Point 8: I do not understand well the phrase of achieved adequate maturity. Please revise.

Response 8: Thank you for your comments and we are really sorry that this part was not clear in our manuscript. We considered replacing the expression with a more specific description to enhance the readability and accuracy of the article. We used more specific expressions such as "reached optimal ripeness for harvesting and winemaking" instead of "achieved adequate maturity". (Line 150)

Point 9: Line 162: Please use the acronym LCFAs if already used in the previous parts of the manuscript.

Response 9: Thank you for your useful suggestion on our manuscript. We have used the abbreviation in line 161 in response to your comment.

Point 10: Line 175 and throughout the manuscript: Please use italics for cis-trans isomerism nomenclature.

Response 10: Thank you for your professional review and we are really sorry for our mistakes. As you suggested, we have scrutinized and revised the entire manuscript to ensure that italics are used throughout to indicate cis-trans nomenclature.

Point 11: Figure 1: Are colored boxes representing the absolute concentration of C6 and C9 volatiles or were the obtained values normalized in some manner given the high-low scale in the legend? Please specify.

Response 11: Thank you for your comments and questions regarding Figure 1. The colored boxes represent the normalized concentration changes in the heat map of product content changes during grapevine berry development in the two plots. Also, we have labeled them in the figure.

Point 12: Lines 247-249: This is incorrect. The majority of the volatile compounds in the grapes of Cabernet Sauvignon were not studied in the present paper as it did not take into the account a plethora of volatile compounds, such as carbonyl compounds, methoxypyrazines, norisoprenoids, terpenes, benzene derivatives and so on. Moreover, the impact of cluster thinning and vintage has been previously studied simultaneously on Cabernet Sauvignon (see https://doi.org/10.1016/j.foodres.2019.03.061), although it is true that authors studied only one vineyard site. Please revise.

Response 12: We appreciate your professional review work on our manuscript and we have adjusted this content.

Point 13: Table 2: It would be more informative for the reader if the authors could add the obtained (mean) values to put the results in the context. The same goes for Table 3.

Response 13: Thank you very much for your detailed review and valuable comments on our paper. After careful consideration, we believe that presenting the full mean data in the current table may make the table too large and complex, which may distract the reader and is not conducive to clearly communicating the results of our study. The current table already succinctly and clearly demonstrates the effects of treatment, vintage, farm, and their interactions on grape composition and wine volatiles, which were the focus of our study. Therefore, based on these considerations, we prefer to keep the current table in its current form.

Point 14: Line 257: Authors did not perform sensory analysis, therefore the statement that the cited ethyl esters contributed to the red-berry aromas is taken from the literature. Please revise.

Response 14: We appreciate your professional review work on our manuscript and we have adjusted the content according to your suggestions. We modified the relevant sentence to read, "According to previous reports in the literature [50], ethyl hexanoate contributes to red-berry aromas together with ethyl esters such as ethyl caprylate, ethyl butyrate, and ethyl 3-hydroxybutyrate." Such a modification would make it clearer that our statement is based on existing research and not on new findings in this study. (Lines 253-256)

Point 15: Lines 340-368: I miss the explanation how the results were expressed in the present study. Were established calibration curves for the quantitation, or are the results expressed as semi-concentration considering the internal standard? Moreover, was the series of n-alkanes used for the calculation of linear retention indices? What was an acceptable mass spectra matching limit that was considered for identification of the compounds? Please be precise.

Response 15: Thank you for your comments and we are really sorry that this part was not clear in our manuscript. We have modified the sections "3.3. GC-MS analysis of C6 volatile compounds in grapes and wines" and "3.1. Reagents and standards" so that we can present our experimental methods more clearly. (Lines 303-304 and lines 366-376)

We established calibration curves for quantitative analysis. And a series of n-alkanes were used to calculate linear retention indices. The retention times of these n-alkanes were used to locate the target compounds in the chromatogram and to calculate their retention indices, thus aiding in compound identification. For compound identification, we set an acceptable limit for mass spectral matching. Specifically, we required a minimum of 85% match of the mass spectra to ensure accurate identification of compounds. In addition, we combine data such as retention index for comprehensive judgment to improve the accuracy of identification.

Reviewer 2 Report

Comments and Suggestions for Authors

Very minor edits, please see attached document.

Author Response

Dear reviewer:

Thank you again for your decision and constructive comments on my manuscript “Influences of cluster thinning on fatty acids and green leaf volatiles production of Cabernet Sauvignon grapes and wines in the northwest of China”. We have carefully considered your suggestion and make changes. The entire manuscript has been done English editing again.

The red part in manuscript that has been revised according to your comments. Revision notes, point-to-point, are given as follows:

Point 1: repetition. please correct

Response 1: We are really sorry for this error. We have made changes in lines 136-137 based on your comments.

Point 2: added a comma and lowercase W

Response 2: Thank you very much for your guidance and correction. We have made changes in line 242 based on your comments.

Point 3: deleted period, lower case w

Response 3: Thank you very much for your guidance and correction. We have made changes in line 245 based on your comments.

Point 4: until

Response 4: We are really sorry for this error. We have made changes in line 404 based on your comments.
